# Influence of Relative Age on Physical Condition and Academic Performance in Adolescents

**DOI:** 10.3390/bs14030181

**Published:** 2024-02-25

**Authors:** Luis Miguel Fernández-Galván, Noelia Belando-Pedreño, Benito Yañez-Araque, Jorge Sánchez-Infante

**Affiliations:** 1Faculty of Sport Science, Jaime I University, 12006 Castellon de la Plana, Spain; luis.fernandez@euseste.es; 2Health and Sport Sciences University School (EUSES), Rovira i Virgili University, 43002 Amposta, Spain; 3Faculty of Physical Activity and Sport Science, European University, 28670 Madrid, Spain; noelia.belando@universidadeuropea.es; 4Department of Business Administration, School of Industrial and Aerospace Engineering, University of Castilla-La Mancha Toledo, 13071 Ciudad Real, Spain; benito.yanez@uclm.es; 5Faculty of Physical Therapy and Nursing, University of Castilla-La Mancha, 45071 Toledo, Spain; 6Faculty of Health Sciences, Universidad Francisco de Vitoria, 28223 Madrid, Spain

**Keywords:** age differences, education, developmental differences, performance

## Abstract

Annual age grouping is a common organizational strategy in academics and sports. This strategy could promote the relative effects of age, which refers to the (dis)advantages that subjects who were born in the first or last months of the year may suffer. The consequences could be minimized, resulting in better physical and/or academic results. The objective of the study was to evaluate the influence of the quarter of birth and examine the correlation between physical condition variables and academic performance. The sample included 79 students (51.90% females) 13.46 ± 0.21 years old in the second year of Obligatory Secondary Education. The physical variables of this study were explosive power, cardiorespiratory capacity, speed, flexibility, and muscle strength. They were evaluated using the horizontal jump tests, Cooper test, 50 m sprint, sit-and-reach test, and medicine ball exercises, respectively. The academic variables were obtained from the average academic grade, grouped by key competencies. An Analysis of Covariance (ANCOVA), controlling for gender, body mass, and height, revealed significant differences between male students born in the first quarter compared to those born in subsequent quarters in all physical condition variables (except for cardiorespiratory capacity). For female students, significant differences were observed only in the explosive power variable. No statistically significant intra-sex differences were evident in academic performance at any time of measurement. Additionally, moderate to large correlations were found in the physical condition and academic variables. It is concluded in relation to the need to implement diverse strategies regarding the process of developing physical conditions in adolescence that satisfy the growth needs of students based on age and sex.

## 1. Introduction

The school organization policy for grouping students in the same academic year is based on the date of birth. In general, students born in the same year are grouped into the same course [1]. This grouping of students in the academic year varies in different countries; thus, in Spain, the grouping takes place from 1 January to 31 December; however, in other countries, such as England, students born between 1 September and 31 August of the following year are grouped in the same course [2]. This difference in age between individuals within the same age group is referred to as relative age, and its consequences are known as the Relative Age Effect (RAE) [3].

These inequalities could lead to (dis)advantages both in academic performance [4] and in physical performance [5,6]. Most of the published research on RAE and its impact on academic performance originates from Norway [7], Britain [8], Belgium [9], or the United States [10]. Despite the variations among these studies, including differences in participant ages, backgrounds, students with special educational needs, and varying sample sizes, a consistent trend emerges: younger students within the school year face more difficulties than relatively older students [11]. This pattern becomes more significant with younger students but reverses upon reaching higher education.

At a sporting and physical performance level, the effects of RAE have been studied since 1985, when Barnsley and Thompson [12] found that 40% of hockey players were born in the first three months of the year. The effect of RAE has a great effect on the physical conditions of children, affecting, for example, the selection of male and female athletes for international competitions [13]. Some physical abilities increase with growth and maturation, such as strength [14] and resistance [15]. Currently, this effect has been shown in other sports, such as basketball [16], handball [17], or soccer, where a recent systematic review showed the effects described above [18]. This means that those born in the first months of the year have more options to play on elite teams [19]. It seems that this advantage in performance is given by the greater maturational growth of those born in the first months of the year, which leads to greater body development that could result in a greater application of force at a higher speed [5]. However, in sports such as gymnastics, an overrepresentation of those born in recent months has been found (contrary to the RAE), since this increase in height and weight could harm the flexibility and speed of rotation of the most mature ones [20].

The literature reviewed focuses on the representation of students in the first months of the year, both in sports and at school. However, there are few studies that have delved into the existence of differences in physical and academic conditions linked to the month of birth. Therefore, the objective of this study was to analyze the influence of the quarter of birth on physical condition and academic performance and examine the relationship between variables. Thus, it was hypothesized that the quarter of birth would influence the level of physical condition and the academic performance of the students; likewise, differences would be observed depending on sex.

## 2. Materials and Methods

### 2.1. Research Design

A quantitative methodology research study was carried out [21], with a descriptive approach (collection, analysis, and presentation of data through quantitative measures) and a correlational (non-experimental relationship between closely linked variables) [22]. For the design of the study and the treatment of data, the ethical guidelines of the Declaration of Helsinki of 1964 (latest revision of Fortaleza, Brazil, 2013) were considered. This study was approved by the ethical committee of Rey Juan Carlos University, Madrid, Spain (code: 3103202314923).

### 2.2. Participants

The sample was made up of 79 adolescent students, of which 41 were females (51.9%) with an average age of 13.24 ± 0.19 and 38 were males (48.1%) with an average age of 13.41 ± 0.24 who were studying the second year of Obligatory Secondary Education (ESO). Non-probabilistic convenience sampling was used [23], organizing the total sample over a four-month period, as proposed by various authors [24,25]. The sample was categorized according to the quarter (Q) of birth: Q_1_ from January to April (n = 27), Q_2_ from May to August (n = 25), and Q_3_ from September to December (n = 27). The sample size was calculated using the G*Power software (version 3.1.4), with a total number of 19 individuals estimated, a statistical power (1−β) of 0.95 for the t test between two dependent means (corresponding pair), an α type error of 0.05, and a large effect size of 0.8 [26].

### 2.3. Sample Selection Criteria

All participants had to be enrolled in the second year of ESO and could not be “repeaters” of that same course, with the aim of not influencing the maturation effect. Another exclusion criterion was the diagnosis of metabolic health problems (e.g., neuroendocrine pathologies), osteoarticular and muscular problems, among others, as well as mental health problems, in order to avoid their influence on the physical condition of the sample.

### 2.4. Procedure

The director of the educational center was contacted by sending a formal letter stating the objectives of the investigation. Once the approval of the educational center was obtained, the informed consents were delivered and signed by the students, their parents, or their legal guardians. Subsequently, the students’ academic grades were accessed and recorded in a database. Likewise, the results of the physical condition tests were collected in that database.

### 2.5. Variables and Measurements

Academic performance. Taking as reference the key specific competencies established by Organic Law 3/2020 modifying the Organic Law of Education 2/2006 (LOMLOE) [27] regulated in Royal Decree 107/2022 for the Valencian Community, the subjects were grouped, in consideration of their connections, into (a) multilingual competence in which the subjects of Spanish, Valencian, and English language were included; (b) mathematical competence and in science, technology and engineering in which the subjects of physics and chemistry, technology and mathematics were included; (c) competence in cultural awareness and expression in which the subject of plastic, visual and audiovisual education was included; and (d) citizen competence in which the subject of geography and history was included.

To assess academic performance, grades ranging from 0 to 10 were collected for the subjects included in each of the four competencies considered, and the average was calculated. These grades were obtained during the second semester of the 2021/2022 academic year.

Physical condition. The physical condition variables were assessed over a 2-week period during scheduled physical education classes (Table 1). All evaluations were conducted outdoors; in cases of inclement weather, assessments took place in the school gymnasium. Prior to the measurement, the same type of warm-up lasting 15 min was carried out, managing the intensity with the Borg scale (RPE, perception of effort) [28]. This initial part of the fitness assessment session consisted of 5 min of continuous running at a moderate intensity (RPE 5–6 out of 10), 5 min of dynamic stretching, and 5 min of progressive sprints (50%, 60%, and 70% of the maximum self-perceived speed). All assessments were carried out twice, following a recovery period of 3 min of rest to ensure complete recovery [29].

### 2.6. Statistical Analysis

Statistical analysis was performed using the free software Jeffrey’s Amazing Statistics Program (JASP), version 0.17.1. The descriptive characteristics of the study sample are presented as means and ± standard deviations. The normality of the variables was checked through the Shapiro-Wilk tests and the Levene test to verify the homoscedasticity of the variances. A one-way ANCOVA was performed to evaluate the effect of the quarter of birth on the physical condition and academic performance variables. The analysis was adjusted for sex, weight, and height. When necessary, Tukey’s post hoc test was carried out to observe differences between means (MD) between the quarters. The ANCOVA effect sizes were expressed with omega squared (ω^2^), considering values of <0.06 (small), 0.06 to 0.014 (medium), and >0.014 (large). Cohen’s d effect size (ES) was calculated using the formula ES = t/√(n) [35]. The interpretation of the ES was 0.2 (irrelevant), 0.2 to 0.6 (small), 0.6 to 1.2 (moderate), 1.2 to 2.0 (large), and >2.0 (very large) [36]. Subsequently, to examine the relationship between variables, the Pearson correlation test was carried out. Correlations were interpreted as small (0.10–0.29), moderate (0.30–0.49), large (0.50–0.69), very large (0.70–0.89), or extremely large (≥0.90) [36]. A 95% confidence interval (CI) was considered, and the significance level was *p* < 0.05.

## 3. Results

The descriptive statistics for the demographic variables of physical condition and academic performance are shown in Table 2.

An ANCOVA adjusted for weight and height was carried out to evaluate the effect of the quarter of birth. Statistically significant differences of large magnitude were observed in the explosive power variable (F _(2.76)_ = 19.28, *p* = <0.001, ω^2^ = 0.32) between Q_1_ and Q_2_ (MD = 0.22 cm 95%CI [0.09, 0.35], d = 4.18, *p* = <0.001) and between Q_1_ and Q_3_ (MD = 0.31 cm 95%CI [0.19, 0.44], d = 6.05, *p* = <0.001). Statistically significant differences of large magnitude were also observed in the muscle strength variables (F_(2.76)_ = 6.96, *p* = <0.001, ω^2^ = 0.13) between Q_1_ and Q_3_ (MD = 0.82 cm, 95%CI [0.29, 1.35], d = 3.71, *p* = <0.001). Statistically significant differences of small magnitude in the cardiorespiratory capacity variables (F_(2.76)_ = 3.63, *p* = 0.030, ω^2^ = 0.06) between Q_1_ and Q_3_ (MD = 251.67 m 95%CI [12.33, 491.01], d = 2.51, *p* = 0.040) and in the speed variable (F_(2.76)_ = 3.41, *p* = 0.040, ω^2^ = 0.06) between Q_1_ and Q_3_ (MD = −0.49 s 95%CI [0.02, 0.21], d = −2.32, *p* = 0.050) (see Figure 1). Regarding the academic performance variables, no statistically significant differences were found in any of the competencies.

Regarding the analysis of the sample by gender (see Table 3), the results show statistically significant differences of large magnitude in males in the explosive power variable (F_(2.35)_ = 17.59, *p* = <0.001, ω^2^ = 0.47) between Q_1_ vs. Q_2_ (MD = 0.34 cm 95%CI [0.16, 0.52], d = 4.61, *p* = <0.001) and Q_1_ vs. Q_3_ (MD = 0.40 cm 95%CI [0.22, 0.57], d = 5.51, *p* = <0.001). Furthermore, statistically significant differences of large magnitude were observed in the speed variable (F_(2.35)_ = 4.57, *p* = 0.020, ω^2^ = 0.16) between Q_1_ and Q_2_ (MD = −0.82 cm 95%CI [−1.49, −0.16], d = 3.02, *p* = 0.010). And statistically significant of large magnitude in the muscle strength variable (F_(2.35)_ = 7.10, *p* = <0.001, ω^2^ = 0.24) between Q_1_ and Q_3_ (MD = 1.16 cm, 95%CI [0.39, 1.92], d = 3.71, *p* = <0.001).

For their part, in females, only statistically significant differences of large magnitude were found in the explosive power variable (F_(2.38)_ = 7.98, *p* = <0.001, ω^2^ = 0.25) between Q_1_ and Q_3_ (MD = 0.23 cm, 95%CI [0.09, 0.38], d = 3.99, *p* = <0.001) (see Table 3). No statistically significant differences were observed in the academic performance variables between males and females. Additionally, significant differences were observed between genders, with females obtaining a better result in the sit-and-reach test in the 3 quarters (all *p* = <0.001). On the other hand, males had a better result in the horizontal jump test, 50 m linear sprint, and medicine ball throw tests in Q_1_ (*p* = <0.001), in the medicine ball throw test in Q_2_ (*p* = 0.03), and in the 50 m linear sprint in Q_3_ (*p* = <0.001).

Finally, a Pearson correlation analysis was carried out, in which correlations from moderate (r = 0.50 to 0.58) to large (r = 0.64 to 0.65) were found between the physical condition variables, except for the variable of flexibility, which presented small correlations with the rest of the variables (Figure 2).

For their part, in the academic performance variables, high correlations are observed between the different competencies (*all r* = >0.715) (Figure 3).

## 4. Discussion

The objective of this study was to analyze the effect of the quarter of birth on the physical condition and academic performance of second-year ESO students, as well as the correlation between variables. The main finding of our study was that there were significant differences in the physical condition variables in the males, specifically in the horizontal jump test between Q_1_ vs. Q_2_ and Q_1_ vs. Q_3_, in the 50 m sprint test between Q_1_ and Q_2_, in the sit-and-reach test between Q_1_ and Q_3_, and in the medicine ball throw test between Q_1_ and Q_3_. However, in females, we only found significant differences in horizontal jumping capacity between Q_1_ and Q_3_. Furthermore, there were no differences in academic performance between the different quarters in the same gender, but there were between-gender differences, finding significant differences in favor of females in mathematical competence in Q_1_, Q_2_, and Q_3_, in awareness competence in Q_2_ and Q_3_, and in multilingual and citizenship skills in Q_3_.

These findings agree with previous research, where it was found that those born in the first months of the year obtained better results in physical condition variables [37]. For example, De la Rubia Riaza and Lorenzo Calvo [16] observed that handball athletes who were born in the first four months of the year had advantages over others due to their maturational development; this meant that the teams were nourished by these males and females to increase the team’s performance. This phenomenon is observed both in team sports [38] and in individual sports [13].

The results obtained show statistically significant differences in the variables of explosive power, speed, and muscle strength in males. A possible explanation would be given by the maturational hypothesis, which maintains that children born in the first months of the year, assuming that growth rates are similar, will show physical and cognitive differences within a period of 12 months that will contribute to having more strength and, consequently, to being more effective in its application and to obtaining better physical results [39]. Furthermore, it must be taken into account that the age of the sample (~13 years) coincides with the approach to the maximum altitude speed, in which the subject experiences structural adaptations due to the direct influence on the metabolic system that is intimately related to the amount of surrounding androgenic hormones [40]. These hormones play an essential role in the processes of muscle glycogen synthesis and hypertrophy, which could contribute to improving the ability to generate strength in its different manifestations [41]. Another explanation for the superior performance of those born in the first quarter is that they could have up to one more year of sporting experience, which is known as the “initial performance advantage” effect [42]. Other authors attribute this superior performance to the “Pygmalion effect,” which describes how the achievements of individuals are a product of the expectations placed on them [43].

However, in the variable to estimate cardiorespiratory capacity, no statistically significant differences were found, despite the fact that it is known that in adolescence, there are changes in metabolic, anatomical, and hematological factors, as well as improvements in running economy [39]. These results agree with another observational study carried out on 69 young soccer players from Portugal [44], as well as with the study of Deprez and Coutts [45], where no significant differences were shown between young elite soccer players. In both cases, the explanation was based on the fact that the sample was made up of selected players and, therefore, the group was homogeneous; in our case, Q1 obtained ~15% better results in the test than Q2 and Q3, which would show the heterogeneity of the sample.

As for females, no statistically significant differences were found in the physical condition variables, except for the horizontal jump test. These results agree with other studies [46] and are expected since females, on average, mature 2 years earlier than males [47]. Therefore, at the age of 13, we could find females who are above the maximum growth speed, and therefore, no statistically significant differences are observed.

On the other hand, our results did not show statistically significant differences in the academic performance variables among students born in different quarters. These findings were unexpected, as previous observations have indicated that students born in the later months of the year tend to have significantly lower average academic grades compared to their peers [7]. In addition, these students are 70–80% more likely to repeat a course [9,48]. Furthermore, in other countries, such as the United Kingdom, where the grouping system is different (1 September to 31 August), significant differences were found [2]. Thus, the study carried out by Cobley and McKenna [2] analyzed 657 students between 11 and 14 years old and showed that older students (born between September and December) obtained higher grades in all subjects than younger students (born between January and August), so there was an overrepresentation in gifted and talented programs. In the Gledhill and Ford [8] study, a cross-sectional analysis was conducted with a sample of 8036 students. The results revealed a statistically significant correlation between the time of birth and educational challenges. Specifically, individuals born in the last term of the academic year (the youngest) demonstrated a greater susceptibility to learning difficulties compared to their counterparts born in the early term (the oldest). These findings are consistent with those of another retrospective observational cohort study involving 2768 students, which found lower academic performance in reading, mathematics, and science. Additionally, significant relationships were identified between time of birth and the rate of diagnosis of specific learning difficulties (10% higher) and the rate of grade repetition (25% higher)” [10]. A possible explanation for our results could be that, although there are differences between students born in different quartiles of the year, these decrease over time and with the experience accumulated at the center, and it is possible that at the age of 13–14 years, this difference is not significant.

Finally, a correlation analysis was carried out between the variables of physical condition and academic performance. The horizontal jump test was observed to be moderately correlated with the 50 m sprint test. These results were expected since it is known that the neuromuscular power of the lower hemisphere correlates with the 5 m and 15 m sprints [49]. Specifically, horizontal jumps are better predictors of the 20 m sprint than vertical jumps [30].

Regarding the relationship between physical condition variables and academic variables, there are data that contain the Cooper test and all academic variables. These data are consistent with other studies that showed that students who scored higher on aerobic fitness tests had higher academic scores, specifically in reading and math, while those who were not aerobically fit scored lower [50]. It seems that when the child exerts himself physically, there is greater serum availability of oxygen and nutrients in the brain, which allows optimal brain function for a longer time [51]. In addition, aerobic exercise improves neural activity and synapses (communication between neurons), which favors concentration during learning [51]. Therefore, aerobic exercise could influence the student’s learning ability [52]. Although these statements are in line with other studies, strategies should be sought to enhance the learning of males and females through physical activity, since the practice of physical activity in adolescents improves their level of learning [53].

Among the limitations of the study is the selection of the sample, which was based on accessibility rather than randomization, thus compromising the external validity of the study. Additionally, the sample size of students is small, which limits the ability to draw scientifically valid deductions. It is important to note that no effort was made to account for potential confounding factors such as cognitive ability, socioeconomic status, or students who may have completed part of their education in different school districts or states. On the other hand, the main strength of this study was the homogeneous sample in terms of age and gender. Subsequent studies might consider creating longitudinal research projects that track students from primary education through secondary education while taking into account potential confounding variables. This approach could lead to more definitive outcomes and provide insight into the idea that differences in academic achievement among students of varying relative ages may initially stem from maturation, with this effect decreasing as maturation becomes less significant. Failing to address this issue could result in unfortunate outcomes for younger students who are biologically and developmentally less mature.

## 5. Conclusions

In conclusion, the results of this study suggest that adolescent students born in the first quarter of the year were associated with better results in physical condition variables than those born in subsequent months. However, no statistically significant differences were observed in academic performance. Our findings underscore the importance of considering the month of birth in the assessment of physical condition tests. Schools might explore the possibility of gradually adjusting the timing, format, and content of students’ physical assessments based on their relative age. One possible solution could be to consider two groups (January to June and July to December).

## Figures and Tables

**Figure 1 behavsci-14-00181-f001:**
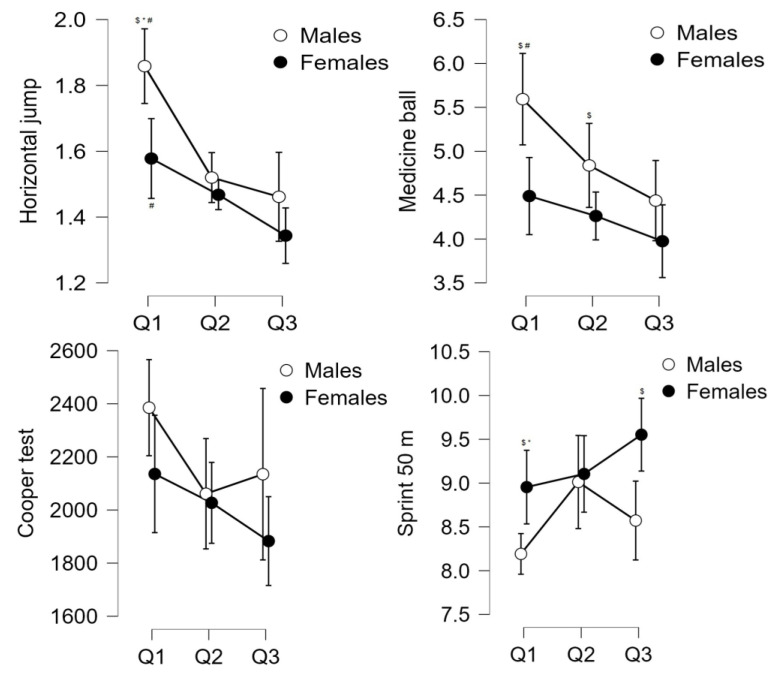
ANCOVA analysis shows the differences between the quarters of birth in the physical condition variables after adjusting for weight and height. Note. Q_1_ (January to April); Q_2_ (May to August); Q_3_ (September to December). ^$^ indicates that it is significant between genders in the same quarter; * indicates that it is significant vs. Q_2_ in the same gender; ^#^ indicates that it is significant vs. Q_3_ in the same gender.

**Figure 2 behavsci-14-00181-f002:**
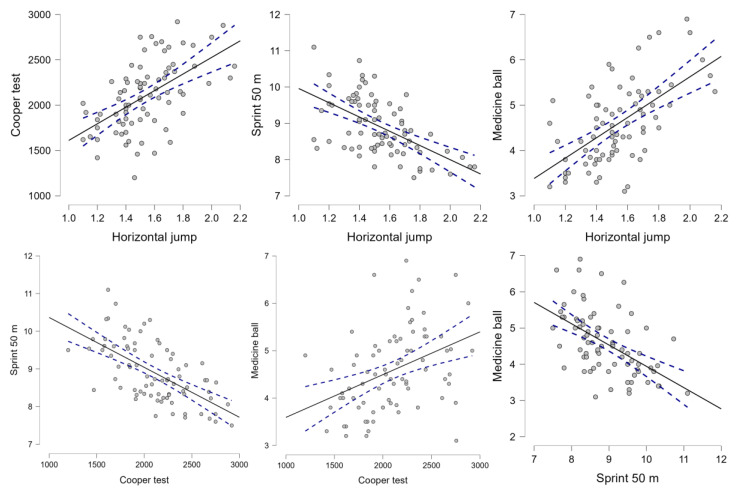
Pearson correlation analysis illustrates the interrelationships among physical condition variables. Individual data points indicate the correlations across different subjects, adjusted for weight and height. The solid line signifies the central tendency, while the dashed lines represent confidence intervals.

**Figure 3 behavsci-14-00181-f003:**
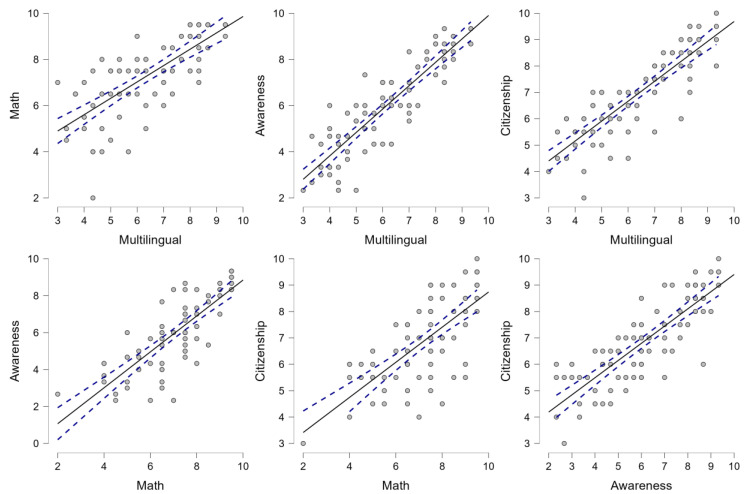
Pearson correlation analysis illustrates the interrelationships among various academic performance measures. Individual data points indicate the correlations across different subjects, adjusted for weight and height. The solid line signifies the central tendency, while the dashed lines represent confidence intervals.

**Table 1 behavsci-14-00181-t001:** Physical condition variables.

Variable	Procedure	Author
Explosive power	The participants performed a horizontal jump; to do so, they stood behind the starting line with their feet shoulder-width apart and bent their knees to an angle of ~120°. To measure the distance jumped, a tape measure oriented perpendicular to the starting line and fixed to the ground was used. The maximum distance in centimeters (cm) achieved by the subject was taken into account.	Maulder and Cronin [30]
Cardiorespiratory capacity	The participants performed the Cooper test on an athletic track, which consisted of running the greatest distance possible in a time of 12 min. The distance traveled in meters (m) was taken into account.	Cooper [31]
Speed	The participants performed the 50 m straight-line sprint test on an athletic track. To measure the time taken, a radar gun placed on a tripod 10 m behind the subject and at a height of ~80 cm was used. The minimum time in seconds (s) taken by the subject was considered.	Di Prampero, Fusi [32]
Flexibility	The participants performed the trunk anterior flexion test (sit-and-reach test). To do this, they took off their shoes and sat on the floor with their legs fully extended and the soles of their feet resting against the box. Then, participants clasped their hands over each other and slowly leaned and stretched forward as far as possible, sliding their fingers along the top of the box and holding the final position for 2–3 s. The maximum distance in cm achieved by the subject was taken into account.	Mayorga-Vega, Merino-Marban [33]
Muscle strength	The participants performed a throw with a 3 kg medicine ball. To do this, they stood behind the starting line, took the ball with both hands, carried it back to gain momentum, and threw it at an angle of ~45° as far as possible. Chalk was used to mark the throw on the ground. The maximum distance in cm achieved by the subject was taken into account.	Gabbett, Georgieff [34]

**Table 2 behavsci-14-00181-t002:** Characteristics of the study sample by quarters.

Variables	Q_1_ (n = 27)	Q_2_ (n = 25)	Q_3_ (n = 27)
Demographics			
Age	13.72 ± 0.15	13.48 ± 0.23	13.18 ± 0.11
Weight (kg)	55.16 ± 7.87	53.89 ± 9.41	54.01 ± 8.78
Height (cm)	160.62 ± 5.11	158.87 ± 7.89	155.89 ± 6.58
Physical condition			
Explosive power (Horizontal jump) (cm)	1.71 ± 0.24 *^#^	1.49 ± 0.10	1.40 ± 0.19
Cardiorespiratory capacity (Cooper) (m)	2255.93 ± 361.67 ^#^	2043.44 ± 284.65	2004.26 ± 435.87
Speed (Sprint 50 m) (s)	8.59 ± 0.70 ^#^	9.06 ± 0.76	9.08 ± 0.87
Flexibility (Sit-and-reach) (cm)	27.37 ± 10.10	23.84 ± 8.48	22.37 ± 11.81
Muscle Strength (Medicine ball) (cm)	5.02 ± 0.97 ^#^	4.54 ± 0.67	4.20 ± 0.76
Academics performance			
Multilingual competence	6.84 ± 1.53	6.13 ± 1.80	5.99 ± 1.95
Mathematical competence	7.80 ± 1.35	7.08 ± 2.03	6.87 ± 1.58
Consciousness competence	6.67 ± 1.98	5.81 ± 2.22	6.02 ± 1.84
Citizenship competence	7.37 ± 1.33	6.40 ± 1.75	6.93 ± 1.55

Note. Data are presented as means and ± standard deviations. Q_1_ (January to April); Q_2_ (May to August); Q_3_ (September to December). * Indicates that it is significant vs. Q_2_. ^#^ Indicates that it is significant vs. Q_3_.

**Table 3 behavsci-14-00181-t003:** Characteristics of the study sample by gender and quarters (n = 79).

	Q_1_ (n = 27)		Q_2_ (n = 25)		Q_3_ (n = 27)	
Variables	Males (n = 13)	Females (n = 14)	Males (n = 12)	Females (n = 13)	Males (n = 13)	Females (n = 14)
Demographics						
Age	13.79 ± 0.41	13.51 ± 0.09	13.41 ± 0.24	13.24 ± 0.19	13.12 ± 0.23	13.09 ± 0.22
Weight (kg)	58.28 ± 5.87	53.24 ± 6.17	55.48 ± 4.17	51.19 ± 3.87	53.28 ± 6.87	50.98 ± 4.87
Height (cm)	159.89 ± 6.87	162.58 ± 5.24	157.89 ± 5.98	159.68 ± 6.58	156.58 ± 4.21	155.65 ± 5.87
Physical condition						
Explosive power (cm)	1.86 ± 0.19 ^$^*^#^	1.58 ± 0.21 ^#^	1.52 ± 0.12	1.47 ± 0.07	1.46 ± 0.22	1.34 ± 0.15
Cardiorespiratory (m)	2385 ± 299	2135 ± 382	2061 ± 534	2027 ± 252	2134 ± 534	1883 ± 289
Speed (s)	8.19 ± 0.38 ^$^*	8.95 ± 0.73	9.01 ± 0.84	9.10 ± 0.72	8.57 ± 0.74 ^$^	9.55 ± 0.72
Flexibility (cm)	23.15 ± 11.65 ^#^	31.29 ± 6.66 ^$^	19.00 ± 5.94	28.31 ± 8.15 ^$^	14.92 ± 6.65	29.29 ± 11.46 ^$^
Muscle strength (cm)	5.59 ± 0.86 ^$#^	4.49 ± 0.76	4.84 ± 0.75 ^$^	4.26 ± 0.45	4.44 ± 0.76	3.98 ± 0.72
Academics performance						
Multilingual	6.82 ± 1.71	6.86 ± 1.40	6.08 ± 1.81	6.18 ± 1.86	5.49 ± 2.01	6.45 ± 1.83 ^$^
Math	7.54 ± 1.27	8.04 ± 1.42 ^$^	6.42 ± 2.36	7.69 ± 1.51 ^$^	6.15 ± 1.52	7.54 ± 1.38 ^$^
Awareness	6.64 ± 2.08	6.69 ± 1.96	5.47 ± 2.27	6.13 ± 2.22 ^$^	5.54 ± 1.86	6.48 ± 1.77 ^$^
Citizenship	7.35 ± 1.45	7.39 ± 1.26	6.17 ± 1.48	6.62 ± 2.01	6.65 ± 1.61	7.18 ± 1.50 ^$^

Note. Data are presented as means and ± standard deviations. Differences between gender and quarter are examined through independent samples (ANCOVA) after adjusting for weight and height. Q_1_ (January to April); Q_2_ (May to August); Q_3_ (September to December). ^$^ indicates that it is significant between genders in the same quarter; * indicates that it is significant vs. Q_2_ in the same gender; ^#^ indicates that it is significant vs. Q_3_ in the same gender.

## Data Availability

All data sets used in this study can be accessed through a request to the author Luis Miguel Fernández-Galván (galvan@uji.es).

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
