# Peer review of "Influence of Relative Age on Physical Condition and Academic Performance in Adolescents"

_behavsci, 2024, doi:10.3390/bs14030181_

Round 1

Reviewer 1 Report

Comments and Suggestions for Authors

The study addresses the issue of age grouping in the organisation of class groups in schools, to reason about the relative effect of age on a number of variables including physical activity and academic performance.

The article deals with an aspect that is often little considered both in the educational field in relation to the analysis of special educational needs, including specific learning disorders, and in the sporting field in the selection of athletes for talent recruitment.

The scientific literature on this specific topic is in fact underdeveloped.

Unfortunately, the sample of students is small, and it does not allow to make scientifically valid deductions but initiates the observation of the phenomenon in an exploratory form.

Methods

The study should be replicated on a larger sample, not only numerically but also more representative of a given population, covering more territorial areas of Spain and not just a sample of a single school.

In addition to the average levels reported in key competences, it would have been interesting to include the variables of academic performance in science and humanities subjects, which would have added further possibilities for reflection with respect to academic performance.

Conclusions

The conclusions do not summarise the results obtained; an analysis of the strengths and weaknesses of the study and future research perspectives in the light of the findings is lacking.

Point 1 ,  2  e 6 -  "Is the content succinctly described and contextualized with respect to previous and present theoretical background and empirical research (if applicable) on the topic? " and  Are all the cited references relevant to the research?-

The theoretical backgroud is good but more recent scientific studies (from 2017 onwards) could be added.

Point 3 - sample: Considering the need to split the sample into quarters, a larger sample would have been useful.

Variables and measurements: there is a need to better describe the values related to the assessment of competence outcomes (numeric or descriptive evaluations, how the areas of competence were grouped)   and to describe the administration procedures of physical test (time schedule)

Point 7 conclusions:describe the study's limitations, strengths and perspectives for future investigations

Author Response

The main changes in the study are shown in the highlighted color. In addition to the PDF document, I add comments to our changes: 

Manuscript ID behavsci-2856836

Dear Editor and reviewer,

Please find enclosed a revision of our manuscript, Influence of relative age on physical condition and academic performance in adolescents”. We would like to thank you for giving us the opportunity to revise and improve our manuscript. We would also like to thank the Senior Associate Editor and Reviewers for their thoughtful and constructive comments which have made the manuscript stronger. Changes to the original manuscript are highlighted in blue font (reviewer 1) and red font (reviewer 2), and an itemized point-by-point response to the reviewers’ comments is presented below.

AUTHORS’ RESPONSE TO REVIEW 1

COMMMENT

The study addresses the issue of age grouping in the organisation of class groups in schools, to reason about the relative effect of age on a number of variables including physical activity and academic performance.

The article deals with an aspect that is often little considered both in the educational field in relation to the analysis of special educational needs, including specific learning disorders, and in the sporting field in the selection of athletes for talent recruitment.

The scientific literature on this specific topic is in fact underdeveloped.

Unfortunately, the sample of students is small, and it does not allow to make scientifically valid deductions but initiates the observation of the phenomenon in an exploratory form.

RESPONSE

We fully agree with your comments. We believe that there is a need for more scientific studies that address these issues, with a larger sample size that allows extrapolation of the data.

COMMMENT

The study should be replicated on a larger sample, not only numerically but also more representative of a given population, covering more territorial areas of Spain and not just a sample of a single school.

In addition to the average levels reported in key competences, it would have been interesting to include the variables of academic performance in science and humanities subjects, which would have added further possibilities for reflection with respect to academic performance.

RESPONSE

Thanks for the comment. With regard to the competences included, we agree with the reviewer's reflection to include those related to the sciences and humanities. In this case, we thought it appropriate to base ourselves on the key competences included in the exit profile of the educational regulations in application in Spain. In this sense, the study has analysed mathematical competence and competence in science, technology, and engineering. However, we have not been able to add the humanities competence because it is not included in the regulations, but we have added related subjects such as Spanish language or plastic education.

COMMMENT

The conclusions do not summarise the results obtained; an analysis of the strengths and weaknesses of the study and future research perspectives in the light of the findings is lacking.

RESPONSE

Thank you for your comments. We have reworded the conclusions section. and added a paragraph at the end of the discussion about the strengths, weaknesses of the study, and possible future research.

COMMMENT

The theoretical background is good but more recent scientific studies (from 2017 onwards) could be added.

RESPONSE

Thank you for your comment. In the introduction we wanted to mention the most cited studies related to the subject. However, we have reviewed the literature and modified several studies to include more current ones.

COMMMENT

Point 1 ,  2  e 6 -  "Is the content succinctly described and contextualized with respect to previous and present theoretical background and empirical research (if applicable) on the topic? " and  Are all the cited references relevant to the research?-

RESPONSE

Thanks for the comment. We have modified the introduction by adding new, more current references and increasing the theoretical content.

COMMMENT

The theoretical backgroud is good but more recent scientific studies (from 2017 onwards) could be added.

RESPONSE

Thanks for the comment. We have added new, more up-to-date references to the manuscript.

COMMMENT

Point 3 - sample: Considering the need to split the sample into quarters, a larger sample would have been useful.

RESPONSE

Thanks for the comment. We agree with your assessment, the calculation showed us that we needed 19 subjects, but we included much more. We believe that it is necessary to carry out a study of this type on a larger number of the population to draw a conclusion that can be extrapolated to a larger population.

COMMMENT

Variables and measurements: there is a need to better describe the values related to the assessment of competence outcomes (numeric or descriptive evaluations, how the areas of competence were grouped) and to describe the administration procedures of physical test (time schedule).

RESPONSE

We agree with your comment. We have reworded the paragraph for better understanding: It now reads as follows: “To assess academic performance, grades ranging from 0 to 10 were collected for the subjects included in each of the four competencies considered, and the average was calculated. These grades were obtained during the second semester of the 2021/2022 academic year”. “The physical condition variables were assessed over a 2-week period during scheduled physical education classes. All evaluations were conducted outdoors; in case of inclement weather, assessments took place in the school gymnasium.”

COMMMENT

Point 7 conclusions: describe the study's limitations, strengths and perspectives for future investigations.

RESPONSE

We fully agree with your comment. We have now rephrased the conclusion and strengths and weaknesses of the study and future research perspectives to make them clearer for the readers. Now it can be read: In conclusion, the results of this study suggest that adolescent students born in the first quarter of the year were associated with better results in physical condition variables than those born in subsequent months. However, no statistically significant differences were observed in academic performance. Our findings underscore the importance of considering the month of birth in the assessment of physical condition tests. Schools might explore the possibility of gradually adjusting the timing, format, and content of students’ physical assessments based on their relative age. One possible solution could be to consider two groups (January to June and July to December).”

Regarding limitations, strengths, and future research lines, we have reworded the paragraph, and it is now readable: “Among the limitations of the study is the selection of the sample, which was based on accessibility rather than randomization, thus compromising the external validity of the study. Additionally, the sample size of students is small, which limits the ability to draw scientifically valid deductions. It's important to note that no effort was made to account for potential confounding factors such as cognitive ability, socioeconomic status, or students who may have completed part of their education in different school districts or states. On the other hand, the main strength of this study was the homogeneous sample in terms of age and gender. Subsequent studies might consider creating longitudinal research projects that track students from primary education through secondary education, while taking into account potential confounding variables. This approach could lead to more definitive outcomes and provide insight into the idea that differences in academic achievement among students of varying relative ages may initially stem from maturation, with this effect decreasing as maturation becomes less significant. Failing to address this issue could result in unfortunate outcomes for younger students who are biologically and developmentally less mature.”

Reviewer 2 Report

Comments and Suggestions for Authors

The most recent scientific literature increasingly emphasizes the need to pay attention to the athlete and their level of physical and cognitive development, going beyond mere chronological age. The Canadian LTAD model aligns with this approach, and I believe your study could further emphasize the holistic view of individuals and the educational system (both in sports and academics) should take this into account. I'm not entirely clear on the decision to also study academic performance. Overall, I must admit I struggled to understand certain parts of your paper. Greater clarity could lead to a better final outcome.

Abstract:

"In general, I believe that writing a clear objective for the study is essential. You have divided it into different parts (lines 22-23, lines 29-30), making it difficult for readers to grasp the true aim or aims of the study."

Introduction:

Main revision:

In general, it's tiring to read the introduction without having to pause several times on its parts to understand them well. It's difficult to find the references that create the theoretical framework for the chosen objective in the study. Let me explain further: typically, in the introduction, we can answer the question "why did the author set out to achieve this particular objective?" Through this introduction, I didn't immediately grasp the motivation behind the choices made (for example the choice to study the sex effect). This problem could be solved by trying to explain what is stated in the literature more precisely and using sentences that are simpler and more understandable to the reader.

Minor revisions:

Lines 48-50:  the sentence “despite this, we can observe differences of up to 3-4 chronological years, without also taking into account possible differences at the maturational level” is not very clear. I think that you have to go into detail of the difference’s’ concept.

Line 52: the Relative Age isn’t a “term”. It might refer to the age of a child in relation to their peers. For example, a child may be considered relatively older if they were born in January compared to a child born in December of the same year, due to differences in school enrollment cutoff dates. think that a clearer and more precise definition of RAE is necessary to improve the understanding of the article.

Line 55-56: I suggest you to implement the bibliography consistently with the study since it considers physical condition and not levels of physical activity.

Methods

Lines 99-101: in the percentage values​​indicated, I don't think it's necessary to use zero.

Lines 126-132: During this period, you make a list of the disciplinary areas. Probably, it would be appropriate to use semicolons and not commas after the description of each area.

Lines 133-135: The grading scale should be specified. For example, do they range from 1 to 10?

Lines 136-137: Reading this sentence, it seems that one day was used for the evaluation of each student. This choice seems complicated to pursue. I assume that evaluations were done in one day for multiple students, perhaps through their division into groups. If that's the case, the procedure used should be explained better.

Lines 143-144: you have considered the longer time, but in the speed test, the shorter time should be considered. This part should be rewritten consistently with the tests used.

Table 1:

-       The Cooper test serves to roughly measure VO2 max (or maximum aerobic power), which is the body's capacity to store, transport, and consume oxygen within a minute during intense physical activity. I'm not convinced that the term "cardiorespiratory capacity" is the most accurate (the same applies to the abstract).

-       The units of measurement used should be included.

-       You mention 'physical condition' here, but in other parts of the text, you refer to 'physical fitness.' Identify the most appropriate term between the two used and use it consistently throughout the text after providing a clear definition in the introduction.

Table 2.

-       The units of measurement used should be included.

The 'discussion' section needs to be revised in light of the new introduction.

It is advisable to review the bibliographical references. Small errors are found in references 2, 8, 15, 22, 34, 36, 44 and 25, such as the presence of additional dashes, the lack of a capital letter at the beginning, the absence of page numbers, the country of publication, etc

Author Response

The main changes in the study are shown in the highlighted color.
In addition to the PDF document, I add comments to our changes: 

Manuscript ID behavsci-2856836

Dear Editor and reviewer,

Please find enclosed a revision of our manuscript, Influence of relative age on physical condition and academic performance in adolescents”. We would like to thank you for giving us the opportunity to revise and improve our manuscript. We would also like to thank the Senior Associate Editor and Reviewers for their thoughtful and constructive comments which have made the manuscript stronger. Changes to the original manuscript are highlighted in blue font (reviewer 1) and red font (reviewer 2), and an itemized point-by-point response to the reviewers’ comments is presented below.

AUTHORS’ RESPONSE TO REVIEW 2

COMMMENT

The most recent scientific literature increasingly emphasizes the need to pay attention to the athlete and their level of physical and cognitive development, going beyond mere chronological age. The Canadian LTAD model aligns with this approach, and I believe your study could further emphasize the holistic view of individuals and the educational system (both in sports and academics) should take this into account. I'm not entirely clear on the decision to also study academic performance. Overall, I must admit I struggled to understand certain parts of your paper. Greater clarity could lead to a better final outcome.

RESPONSE

We fully agree with your comments. We are familiar with the Youth Physical Development Model proposed by Lloyd et al. (2012). This proposal is based on physical aspects and their influence on sports performance. In this sense, it is observed that a very high percentage analyse this effect on sport and its derivatives (identification of talent), so we wanted to analyse how it could influence academic performance. This is the main justification for the present study.

COMMMENT

Abstract. "In general, I believe that writing a clear objective for the study is essential. You have divided it into different parts (lines 22-23, lines 29-30), making it difficult for readers to grasp the true aim or aims of the study."

RESPONSE

We agree with your comment, we have reworded the abstract and it is now readable: “… The academic variables were obtained from the average academic grade grouped by key competencies. An Analysis of Covariance (ANCOVA), controlling for gender, body mass, and height, revealed significant differences between male students born in the first quarter compared to those born in subsequent quarters in all physical condition variables (except for cardiorespiratory capacity). For female students, significant differences were observed only in the explosive power variable…”

COMMMENT

Introduction: (Main revision):

In general, it's tiring to read the introduction without having to pause several times on its parts to understand them well. It's difficult to find the references that create the theoretical framework for the chosen objective in the study. Let me explain further: typically, in the introduction, we can answer the question "why did the author set out to achieve this particular objective?" Through this introduction, I didn't immediately grasp the motivation behind the choices made (for example the choice to study the sex effect). This problem could be solved by trying to explain what is stated in the literature more precisely and using sentences that are simpler and more understandable to the reader.

RESPONSE

Thank you for your comment. We believe you are right, so we have revised and reworded the introduction to make it clearer for the reader. In response to your comments, we have made the introduction more precise in addressing a theoretical framework and using simpler and more understandable sentences.

COMMMENT

Lines 48-50:  the sentence “despite this, we can observe differences of up to 3-4 chronological years, without also taking into account possible differences at the maturational level” is not very clear. I think that you have to go into detail of the difference’s’ concept.

RESPONSE

Thank you for your input. We have reviewed the sentence and have decided to delete it as it does not provide any more information than the above.

COMMMENT

Line 52: the Relative Age isn’t a “term”. It might refer to the age of a child in relation to their peers. For example, a child may be considered relatively older if they were born in January compared to a child born in December of the same year, due to differences in school enrollment cutoff dates. think that a clearer and more precise definition of RAE is necessary to improve the understanding of the article.

RESPONSE

We agree with your statement, which is why we have reworded the paragraph and it can now be read: “…This grouping of students in the academic year varies in different countries, thus, in Spain the grouping takes place from January 1st to December 31st, however, in other countries such as England, students born between September 1st and August 31st of the following year, are grouped in the same course [2]. This difference in age between individuals within the same age group in referred to as relative age, and its consequences are known as the Relative Age Effect (RAE)…”

COMMMENT

Line 55-56: I suggest you to implement the bibliography consistently with the study since it considers physical condition and not levels of physical activity.

RESPONSE

Your statement is correct, which is why we have deleted the sentence and the bibliographic reference.

COMMMENT

Lines 99-101: in the percentage values ​​indicated, I don't think it's necessary to use zero.

RESPONSE

Thank you for your comment. We have eliminated the zero.

COMMMENT

Lines 126-132: During this period, you make a list of the disciplinary areas. Probably, it would be appropriate to use semicolons and not commas after the description of each area.

RESPONSE

Ok, we have modified this.

COMMMENT

Lines 133-135: The grading scale should be specified. For example, do they range from 1 to 10?

RESPONSE

and now it can be read:

COMMMENT

Lines 136-137: Reading this sentence, it seems that one day was used for the evaluation of each student. This choice seems complicated to pursue. I assume that evaluations were done in one day for multiple students, perhaps through their division into groups. If that's the case, the procedure used should be explained better.

RESPONSE

Thank you for the feedback, it's an error. We have reformulated the sentence, and now it can be read ”The physical condition variables were assessed over a 2-week period during scheduled physical education classes. All evaluations were conducted outdoors; in case of inclement weather, assessments took place in the school gymnasium.”

COMMMENT

Lines 143-144: you have considered the longer time, but in the speed test, the shorter time should be considered. This part should be rewritten consistently with the tests used.

RESPONSE

COMMMENT

Table 1:

-       The Cooper test serves to roughly measure VO2 max (or maximum aerobic power), which is the body's capacity to store, transport, and consume oxygen within a minute during intense physical activity. I'm not convinced that the term "cardiorespiratory capacity" is the most accurate (the same applies to the abstract).

-       The units of measurement used should be included.

-       You mention 'physical condition' here, but in other parts of the text, you refer to 'physical fitness.' Identify the most appropriate term between the two used and use it consistently throughout the text after providing a clear definition in the introduction.

RESPONSE

Thank you for your comments. We have reviewed the bibliography and, indeed, the term "cardiorespiratory capacity" is the most appropriate for our study, although it is true that we have found other authors who call it: "endurance capacity" or "cardiovascular resistance".

- The units of measurement have been specified in Table 1 and have been added to the rest of the tables.

- Finally, we have thoroughly reviewed the entire text of the study and have consistently added the term "physical fitness".

Tapking, C., Popp, D., Herndon, D. N., Branski, L. K., Mlcak, R. P., & Suman, O. E. (2018). Estimated versus achieved maximal oxygen consumption in severely burned children maximal oxygen consumption in burned children. Burns : journal of the International Society for Burn Injuries, 44(8), 2026-2033. https://doi.org/10.1016/j.burns.2018.06.004

Alvero-Cruz J.R., Giráldez García M.A., Carnero E.A.. Reliability and accuracy of Cooper's test in male long distance runners. Rev Andal Med Sport [Internet]. 2017 [cited 2024 Feb 16]; 10(2): 60-63. Disponible en: http://scielo.isciii.es/scielo.php?script=sci_arttext&pid=S1888-75462017000200060&lng=es. https://dx.doi.org/10.1016/j.ramd.2016.03.001.

Sánchez Rojas, I. A. . (2021). Correlational analysis of the validity and reliability of the Cooper Test versus conventional field tests, for the establishment of cardiovascular endurance. Impetus Magazine, 11(2), 9-16. https://doi.org/10.22579/20114680.430.

COMMMENT

Table 2.

-       The units of measurement used should be included.

RESPONSE

The units of measurement have been specified in Table 1 and added to the rest of the tables.

COMMMENT

The 'discussion' section needs to be revised in light of the new introduction.

RESPONSE

Thank you for your comment. We have reworded the entire discussion section based on the introduction.

COMMMENT

It is advisable to review the bibliographical references. Small errors are found in references 2, 8, 15, 22, 34, 36, 44 and 25, such as the presence of additional dashes, the lack of a capital letter at the beginning, the absence of page numbers, the country of publication, etc

RESPONSE

Thank you for your comment. We have corrected it according to your indications and reviewed all the bibliography to avoid any errors.

Round 2

Reviewer 2 Report

Comments and Suggestions for Authors

I wanted to express my appreciation for the effort you've put into revising the manuscript. After carefully reviewing the changes you've made in response to my suggestions, I'm pleased to say that I have no further recommendations to offer.